# Chemical and Electrochemical Reductions of Monoiminoacenaphthenes

**DOI:** 10.3390/ijms24108667

**Published:** 2023-05-12

**Authors:** Vera V. Khrizanforova, Robert R. Fayzullin, Tatiana P. Gerasimova, Mikhail N. Khrizanforov, Almaz A. Zagidullin, Daut R. Islamov, Anton N. Lukoyanov, Yulia H. Budnikova

**Affiliations:** 1Arbuzov Institute of Organic and Physical Chemistry, FRC Kazan Scientific Center, Russian Academy of Sciences, 8 Arbuzov Street, 420088 Kazan, Russia; khrizanforovavera@yandex.ru (V.V.K.); robert.fayzullin@gmail.com (R.R.F.); tatyanagr@gmail.com (T.P.G.); khrizanforov@gmail.com (M.N.K.); daut1989@mail.ru (D.R.I.); yulia@iopc.ru (Y.H.B.); 2A.M. Butlerov Institute of Chemistry, Kazan Federal University, 18 Kremlevskaya Street, 420008 Kazan, Russia; 3G.A. Razuvaev Institute of Organometallic Chemistry, Russian Academy of Sciences, 49 Tropinin Street, 603137 Nizhny Novgorod, Russia; anton_lukoyanov@mail.ru

**Keywords:** monoimonoacenaphthene, electrochemical reduction, electrochemical gap, sodium complex, one-electron reduction, two-electron reduction, anion radical, X-ray structure

## Abstract

Redox properties of monoiminoacenaphthenes (MIANs) were studied using various electrochemical techniques. The potential values obtained were used for calculating the electrochemical gap value and corresponding frontier orbital difference energy. The first-peak-potential reduction of the MIANs was performed. As a result of controlled potential electrolysis, two-electron one-proton addition products were obtained. Additionally, the MIANs were exposed to one-electron chemical reduction by sodium and NaBH_4_. Structures of three new sodium complexes, three products of electrochemical reduction, and one product of the reduction by NaBH_4_ were studied using single-crystal X-ray diffraction. The MIANs reduced electrochemically by NaBH_4_ represent salts, in which the protonated MIAN skeleton acts as an anion and Bu_4_N^+^ or Na^+^ as a cation. In the case of sodium complexes, the anion radicals of MIANs are coordinated with sodium cations into tetranuclear complexes. The photophysical and electrochemical properties of all reduced MIAN products, as well as neutral forms, were studied both experimentally and quantum-chemically.

## 1. Introduction

Monoiminoacenaphthenes (MIANs) represent a class of redox-active ligands that are intermediate between acenaphthenequinone (AQ) and acenaphthene-1,2-diimines. Correspondingly, the MIANs steric properties and redox activity are intermediate in this series, suggesting the formation of a new type of complexes with unpredictable reactivity [1,2]. MIANs are strong coordinating ligands [3,4,5,6] because each includes carbonyl and imine functional groups conjugated with the naphthalene fragment.

It is known that the one- or two-electron reduction/oxidation stage is essential in many catalytic reactions, which leads to the formation of a catalytically active species [7,8,9,10,11]. The complexes of non-transition metals with redox-active ligands, obtained by chemical reduction, are widely used in catalysis and small molecule activation [12,13,14,15,16,17,18,19]. From this point of view, bis-iminoacenaphthene (BIAN) complexes have been studied quite extensively. In turn, the MIAN complexes have been far less studied (Figure 1). Thus, it was found that zinc(II) complexes with MIANs are precatalysts for the reaction of the formation of guanidine and urea derivatives [6]. Mn^I^-MIAN complexes exhibited unusually high sensitivity to visible light, even in the solid state, and rapidly release carbon monoxide (CO) upon illumination [20]. Gallium complexes with dpp-MIAN showed moderate catalytic activity for the hydroamination of unsaturated organic compounds [21]. Ruthenium complexes with MIAN exhibited multi-catalytic properties with respect to the reactions of epoxidation of alkenes, oxidation of alcohols, and electrocatalytic oxidation of water [22]. High catalytic activity in the oxidation of alkanes and alcohols with peroxides was found for vanadium complexes of MIANs [18]. Dianionic calcium complexes were found to be active in the reaction of L-lactide polymerization [17].

Electrochemically, MIANs also take an intermediate position between initial AQ and bis-iminoacenaphthene (BIAN), which is evidenced by comparing their redox potentials. This idea was exemplified by a series of compounds: AQ, dpp-MIAN, and dpp-BIAN [16]. However, unlike BIANs [23,24,25,26], the electrochemical properties of MIANs have been studied extremely poorly. In a recent article [16], the one-electron reduction of dpp-MIAN (dpp = diisopropylphenyl) by sodium and potassium metals was shown. In situ EPR-spectroelectrochemical detection of dpp-MIAN anion radical was discussed.

The target objects of this study are a series of MIANs. Their electrochemical properties were studied using voltammetry techniques, and the influence of substituents on the redox properties of the compounds was further evaluated. In this study, frontier orbital difference energies (that is, between the highest occupied molecular orbital (HOMO) and the lowest unoccupied molecular orbital (LUMO)) of MIANs were estimated in four independent ways: experimentally from (i) electrochemical data, ΔE_el_, and (ii) UV/Vis absorption spectra, ΔE_abs_, as well as theoretically with the use of (iii) density-functional theory (DFT), ΔE_DFT_, and (iv) time-dependent density-functional theory (TDDFT), ΔE_TDDFT_, computations. In order to reveal the importance of one- or two-electron transfer reactions for catalysis, we describe the one-electron chemical reduction of MIANs with sodium metal, their electrochemical reduction via controlled potential electrolysis at the first peak potential estimated from cyclic voltammetry (CV), and chemical reduction by NaBH_4_. The products of the three different types of reduction reactions were characterized by single-crystal X-ray diffraction (XRD), and their photophysical and electrochemical properties were additionally studied. The differences between the reduction methods were found. The results obtained could be useful in choosing reducing conditions for designing new catalytic systems requiring the transfer of one or two electrons.

## 2. Results and Discussion

### 2.1. Synthesis and Structure of MIANs

The MIANs were synthesized by coupling one equivalent of AQ with one equivalent of a relevant aniline in the presence of a catalytic amount of formic acid, which led to obtaining the target compounds with a moderate yield (Appendix A). All studied MIANs are known in the literature. However, the crystal structure of most of them is unknown. In our work, as a result of multiple crystallization of the MIANs from saturated solutions of DCM (dichloromethane), we succeeded in obtaining crystals suitable for XRD for **II**, **III**, and **IX** (Figure 2). The C–N, C–C, and C–O distances for **II**, **III**, and **IX** are about 1.27 Å, 1.55 Å, and 1.21 Å, respectively. For redox-active ligands, including MIANs, the lengths of the key C–C, C–N, and C–O bonds of the (initially) –N=C–C=O moiety make a valuable contribution to the identification of the ligand and metal oxidation numbers in complexes.

### 2.2. Redox Properties of MIANs, Theoretical vs. Electrochemical HOMO/LUMO Gaps

Oxidation-reduction potential is a measure of the ability to accept/donate an electron in a solution medium on the electrode surface. One of the most effective electro-analytical techniques to study electroactive species is CV. The redox properties of MIANs were studied using the techniques of CV and differential pulse voltammetry (DPV) in 0.1 M of Bu_4_NBF_4_ solution (supporting electrolyte) in acetonitrile (Figure 1 and Appendix A). Potentials measured by CV and DPV were obtained according to the internal standard, that is, the redox ferrocenium/ferrocene pair (Fc^+^/Fc). Glass carbon surface was used as a working electrode, while a platinum wire was used as an auxiliary electrode. All MIANs were characterized by similar electrochemical behavior. Three reduction peaks were observed for the MIANs in the available cathode area of potentials. ΔE (difference between E_a_ (anodic) and E_c_ (cathodic)) for the peaks lies within the range of 60–110 mV, which indicates the electrochemical reversibility of the electron transfer within the ligand. All reduction peaks are diffusion-controlled, as evidenced by the linear dependence of the root on the scanning rate and the current (see Appendix A). In the anodic potential region, the first oxidation peak corresponding to forming the cation radical lies within the potential range of 0.89–1.29 V (Table 1) and is irreversible. Introducing methyl- (**IV**), ethyl- (**V**), or *iso*-propyl- (**VI**) substituents into positions 2 and 6 of the aryl ring leads to the negative shifting of reduction potential by 100–170 mV as compared to ligand **I**. A similar shift in reduction potentials into the cathodic region is observed for compounds **II** and **III** with Me or OMe substituent in the *para*-position of the aryl ring. In these cases, the potential is shifted by 120–140 mV relative to **I**. Substituting the benzene fragment with naphthyl (**VIII**) or quinolinyl (**IX**) does not practically affect the reduction potentials, although minor changes in potential values are observed.

Different HOMO–LUMO energies of MIANs were estimated in four independent ways: experimentally from (i) electrochemical data, ΔE_el_, and (ii) UV/Vis absorption spectra, ΔE_abs_, as well as theoretically applying (iii) DFT, ΔE_DFT_, and (iv) TDDFT and ΔE_TDDFT_, computations.

First, the HOMO and LUMO energies were calculated quantum-chemically with the use of hybrid PBE0 functional [27] and def-TZVP basis sets [28] (see Appendix A). The energies of HOMO and LUMO, as well as the energy gaps calculated for the gas-phase-optimized structures, are presented in Table 2 and Appendix A.

Previously, Hasan et al. [22] studied the redox properties of neutral BIANs using CV and DFT calculations of frontier orbital energy values. In that paper, there are data for BIANs with phenyl-, mesityl-, and *para*-methoxyphenyl-substituents at the nitrogen atoms. The electrochemical gap between the first oxidation and reduction potentials for these BIANs (based on CV data) decreases from 2.75 through 2.35 to 2.12 eV in the series of phenyl-, mesityl-, and *para*-methoxyphenyl-BIANs, while ΔE_DFT_ in the same series decreases from 3.79 through 3.49 to 3.27 eV. For the similar MIANs, there are the same regularities; thus, in series **I**, **VII**, and **III**, ΔE_el_ as well as ΔE_DFT_ decrease regularly from 2.48 through 2.37 to 2.26 eV and from 4.07 through 3.72 to 3.57 eV, respectively.

Another way to estimate the HOMO–LUMO gap is based on the analysis of the wavelength of the lowest energy absorption band, which, as a rule, is caused by transitions between the frontier orbitals. Thus, for all the studied compounds, the UV/Vis spectra were registered and vertical shifts for the gas-phase optimized structures were calculated quantum-chemically (Appendix A). The predicted spectra agree quite well with the experimental curves (Figure 2). According to the computations, the low energy bands (430–483 nm) correspond to the transition from HOMO to LUMO, the former being located mostly at the aryl substituent and the latter being contributed by the MIAN moiety (see Appendix A).

It should be mentioned that the MIAN contribution to the HOMO depends on the dihedral angle between planes of MIAN and aryl moieties. The latter defines the part of the intramolecular charge transfer (ICT). For the molecules with the almost orthogonal location of MIAN and aryl moieties, MIAN also contributes to HOMO, while the ICT part is lower in the HOMO–LUMO transitions and the corresponding low energy band is weak (as with **IV**, **V**, and **VII**). A decrease in the dihedral CNCC angle leads to the intensification of this band due to its more pronounced ICT character. Analysis of the computed and experimental spectra shows that the introduction of substituents into the aryl group at the nitrogen atom bathochromically shifts the low-energy band. The maximal shift (52 nm) compared to **I** is predicted for **VIII**.

It is known that the first oxidation and/or reduction potential of redox-active matter directly correlates with the energy of the frontier orbitals HOMO and LUMO [24,29,30]. To compute the energies of frontier orbitals HOMO and LUMO directly from the electrochemical data obtained experimentally, Formula (1) can be used, in which E_CV_ is the potential of a half wavelength or peak potential obtained using cyclic voltammetry in the case of reversible systems. In the case of irreversible systems, the HOMO and LUMO energy levels are computed using Formula (2), in which E_DPV_ is the DPV potential of the first oxidation/reduction peak for a substance analyzed.
E_HOMO_ = 4.8 + E_CVox(vs. Fc+/Fc)_ [eV] and E_LUMO_ = 4.8 + E_CVred(vs. Fc+/Fc)_ [eV](1)
E_HOMO_ = 4.8 + E_DPVox(vs. Fc+/Fc)_ [eV] andE_LUMO_ = 4.8 + E_DPVred(vs. Fc+/Fc)_ [eV](2)

Thus, the potential values obtained using the CV and DPV techniques for MIANs **I**-**IX** can be directly used to compute the energy levels of HOMO and LUMO, while the value of the electrochemical gap (difference between the first oxidation peak and the first reduction peak) correlates with the energy value ΔE_el_ = |E_HOMO_ − E_LUMO_| and is shown in Table 1. For the MIANs studied, E_HOMO_ ranges from −5.65 to −5.80 eV, E_LUMO_ ranges from −3.20 to −3.44 eV, and ΔE_el_ for MIANs falls into the range of 2.26–2.56 eV.

Thus, the analysis of ΔE values using different techniques indicates that the general trends or regularities obtained by all the methods, i.e., DFT, TDDFT, electrochemistry, and UV/Vis spectroscopy, intercorrelate well (see Appendix A). Comparability of the results obtained by several independent techniques allows us to estimate the identity of the computation model chosen for this class of compounds.

### 2.3. One-Electron Reduction of Studied MIANs by Sodium Metal

To obtain the anion radicals of MIANs, the one-electron reduction of neutral MIAN by one equivalent of alkali metal was performed (Figure 3). This technique is used quite frequently to obtain various anion radicals of MIANs and BIANs and to further use them for synthesizing other metal complexes. For dpp-BIAN, for instance, complexes with the anion, dianion, trianion, and tetraanion radicals of di-imine were obtained by reducing with an alkali metal (sodium) and characterized structurally [25]. The structures of dpp-MIAN, sodium, and potassium complexes with ligands in the anion radical state were confirmed by XRD [16].

The one-electron reduction of MIANs **I**–**IX** with sodium was carried out in tetrahydrofuran (THF) at room temperature. As a result, sodium complexes **1**–**9**-Na with anionic MIANs were obtained. In the EPR spectra, there is an expected signal with a g-factor of 2.00 and ΔH = 2 G, as well as with a hyperfine interaction constant from the atom of nitrogen and oxygen. Crystal structures of some new sodium complexes were established using XRD (Figure 3). Crystals suitable for XRD study were prepared by the slow diffusion of hexane into a THF solution of the respective sodium complex at −35 °C. Although three studied compounds **5**-Na, **6**-Na, and **7**-Na crystallize in different space groups, the molecular tetrametallic complexes are characterized by a similar structure in the crystal (Figure 3). In each, a Na_4_O_4_ core of a distorted cuboid shape can be distinguished; the sodium ion coordinates the oxygen atom of the THF molecule, the nitrogen atom of the ligand (anion radical), and the three oxygen atoms of the three ligands.

It is interesting to analyze the structure of innocent ligands, especially the distances within –N=C–C=O fragment as a function of the oxidation state of the ligand. Fedushkin et al. showed for dpp-BIAN [25] that the reduction of neutral molecules to the anion radical, dianion, etc., is accompanied by the typical changes in bond lengths in the ligand, namely the characteristic elongation of the C–C bond and shortening of the C–N bond. Ragaini [23] and Khusniyarov [26] found correlations between the bond lengths of a non-innocent ligand and its charge. In the work by Ragaini, the structures of over 300 compounds were analyzed, containing neutral, anion radical, and dianion Ar-BIANs. A clear dependence of the ligand charge on the lengths of the C–C and C–N bonds has been revealed.

For MIANs, as well as for BIANs, the lengths of the C–C, C–N, and C–O bonds of the (initially) –N=C–C=O moiety could be the identifier of the ligand oxidation number. For neutral MIANs synthesized in this study and those previously published by other teams, the C–N, C–C, and C–O distances are ca. 1.27 Å, 1.55 Å, and 1.21 Å, respectively. The DFT-predicted bond lengths agree well with the experimental values. The calculated C–N interatomic distance is ca. 1.26 Å, the C–C interatomic distance is ca.1.54 Å, and the C–O interatomic distance is ca.1.20 Å. For sodium complexes with an anionic state of the ligand, we observed the expected elongation of the C–O (1.277–1.284 Å) and C–N (1.275–1.358 Å) interatomic distances and the shortening of the C–C (1.419–1.459 Å) interatomic distances in the crystals. Thus, changes in the redox-sensitive bond parameters involving –N=C–C=O fragment in the crystals confirm the reduced form of the ligand. There is also a very good agreement with theoretical calculations (C–O distance is ca. 1.23 Å, C–N distance is ca. 1.32 Å, and C–C distance is ca. 1.47 Å). For dpp-MIAN sodium complexes in both dimeric [16] and tetrameric forms, there are no differences in bond lengths.

### 2.4. Photophysical Properties of Sodium Anion Radicals

One-electron reduction of studied MIANs by metallic sodium leads to a change in the sample color from orangish yellow to dark purple, which should result in changes in their UV/Vis spectra. The spectra of **1**–**9**-Na contain a new absorption band at 520–613 nm (Table 3, Appendix A). For calculations, anion radicals **1**–**9**-DFT were considered as negatively charged ions without counter-ions. The experimental and predicted long wave bands of **1**–**9**-Na are batochromocally shifted on 57–110 nm in comparison to corresponding bands of neutral MIANs **I**–**IX** (Table 2).

The DFT bands at 500–650 nm are related to the S_0_-S_2_ transitions (Table 3). For the radicals **8**-DFT and **9**-DFT, the wavelengths corresponding to the S_0_-S_3_ transitions are also presented in Table 3 since they appear to be much more intensive compared to the S_0_-S_2_ one. Analysis of the frontier orbitals contributing to these transitions allows us to divide the ions into three groups (Figure 4): (i) without substituents or with substituents in the *ortho*-position of the aryl group, which is sterically unhindered (**1**–**3**-DFT), (ii) with sterically hindered aryl groups at the N atom (**5**–**7**-DFT), and (iii) with conjugated aromatic groups at the N atom. For the first group, the absorption in the respective range of spectra is related to π–π* transitions in the whole ion; for (ii), the absorption is caused by π–π* transitions within the MIAN part of the ions; and for (iii), computations predict intramolecular charge transfer (Figure 4). All predicted trends are in good agreement with experimental observance. The revealed difference in the frontier orbital location for anion radicals may lead to different reactivity a catalytic activity of their complexes with non-transition metals. These results can be further used for variation of reactivity of complexes by the tuning of ligands.

### 2.5. Preparative Electrochemical Reduction of MIANs

The first reversible reduction peak observed in voltammograms for MIANs **I**–**IX** indicates that reduction provided a stable reduced state in the solution. In our previous work, we showed that the in situ electrolysis of dpp-MIAN at the first reduction peak potential provided paramagnetic anion radicals. In the EPR spectrum, the signal corresponding to one-electron-reduced dpp-MIAN was observed similar to the sodium complex of dpp-MIAN [16].

On the other hand, as mentioned above, the redox properties of BIANs and MIANs are rather close. Previously, Ragaini et al. studied the redox properties of unhindered BIANs in detail [23]. They proposed a two-electron reduction of BIANs, forming BIANH_2_ under electrochemical reduction in the presence of protons. Similar two-electron and two-proton addition to neutral BIANs was observed in reaction with NaBH_4_ as a reducing agent [24].

For the MIANs studied, in the conditions of the preparative controlled potential electrolysis at the first CV reduction potential, 2 F of electricity was found to be required for each mole of the MIAN. No EPR signal was observed for aliquots of the reaction mixture after electrolysis, which confirmed the diamagnetic nature of the reduction product. After extraction of the product by diethyl ether or dimethoxyethane and storing the solution at −35 °C for a few days, crystals suitable for the XRD analysis were obtained.

These compounds represent salts, in which protonated MIAN^−^ is the anion, while Bu_4_N^+^ of the background electrolyte acts as the counter ion (Figure 5). An interesting feature of the crystal structure of **4**-NBu_4_ is that this organic salt is a co-crystal of the initial MIAN **IV** and **4**-NBu_4_ itself. The neutral (**IV**) and anionic (**4**^−^) forms can occupy the same crystal site, which manifests itself in substitution disorder. This leads to the fact that the observed crystal structure turns out to be a superposition of two very close forms, and each MIAN fragment in crystal packing carries an apparent fractional charge of −0.5e.

Key lengths of the C–C, C–N, and C–O bonds within the (initial) –N=C–C=O fragment change regularly in the crystal structure for **1**-NBu_4_ and **3**-NBu_4_. The C–C bond becomes shorter (ca. 1.399 and 1.397 Å for **1**-NBu_4_ and **3**-NBu_4_, respectively), while the C–O and C–N bonds become longer (ca. 1.275 and 1.413 Å for **1**-NBu_4_; 1.281 and 1.411 Å for **3**-NBu_4_, respectively).

Close structural changes in the bond distances were shown for BIANH_2_ [24]. The reduction of *para*-Tol-BIAN by two equivalents of NaBH_4_ leads to dianionic BIANH_2_. Herein, we provided a similar reduction of **V** with NaBH_4_ as a reducing agent. As a result, the new sodium salt **5**-NaBH_4_ was isolated. The crystals suitable for the XRD study were grown from saturated THF solution at room temperature. Salt **5**-NaBH_4_ is characterized by a Na_4_O_4_ core of a distorted cuboid shape (Figure 6). Similar to compounds **5**-Na, **6**-Na, and **7**-Na, the sodium ion coordinates the oxygen atom of the THF molecule, the nitrogen atom of the ligand, and the three oxygen atoms of the three ligands. According to XRD, in contrast to **5**-Na, each N-atom of both ligands in **5**-NaBH_4_ is bonded to an H atom, which was confirmed by difference Fourier maps. The C–C, C–N, and C–O bonds within the –N=C–C=O fragment are 1.378, 1.448, and 1.298 Å, respectively, and this agrees with bond distances for **1**-NBu_4_, **3**-NBu_4_, and **4**-NBu_4._

Interestingly, **5**-NaBH_4_ in the presence of an excess of supporting electrolyte transformed to **5**-NBu_4_, which was confirmed by UV/Vis spectroscopy. This means that the electrochemical reduction of MIANs under electrolysis leads to the salt with Bu_4_N^+^ or sodium counter-ion as a result of two-electron and one-proton reduction following Figure 4.

### 2.6. Comparison of Reduction Potentials for Compounds 1–9-X (Where X = Na or NBu_4_)

The reduction potentials of sodium complexes **1**–**9**-Na and tetrabutylammonium salts **1–9**-NBu_4_ were studied with CV and DPV in 0.1 M solution of Bu_4_NPF_6_ in THF (Figure 7). Sodium complexes **1**–**9**-Na show one quasi-reversible reduction peak at the potentials from −1.57 to −1.81 V. For **1**–**9**-NBu_4_, an irreversible reduction peak at potentials from −2.20 to −2.49 V was observed. In all cases, sodium complexes are more easily reduced than corresponding ammonium salts (Table 4). The difference between the reduction potentials of sodium complexes and ammonium salts is 440–910 mV. Similar trends were observed in the comparison of values of absorbance wavelength for **1**–**9**-X (Table 3 and Table 4). For **1**–**9**-NBu_4_, the long wave absorption band is bathochromically shifted on 15–70 nm, compared to that for **1**–**9**-Na.

## 3. Materials and Methods

The general information, characterization data, experimental procedures, computational details as well as electrochemical, EPR, and magnetic measurements data are available in the Appendix A.

Deposition numbers CCDC 2253912-2253916 and 2254012-2254015 (deposited on 4 April 2023) and 2256110 (deposited on 13 April 2023) contain the supplementary crystallographic data for this paper. These data are provided free of charge by the joint Cambridge Crystallographic Data Centre and FachinformationszentrumKarlsruhe Access Structures service www.ccdc.cam.ac.uk/structures.

## 4. Conclusions

This study provides a detailed analysis of the redox properties of MIANs using various electrochemical techniques. The HOMO–LUMO gap was computed for the MIANs experimentally, based on the data of the oxidation/reduction potentials as well as of the electronic absorption spectra, and theoretically within the DFT and TDDFT approaches. It was found that all methods intercorrelate well. We describe the one-electron chemical reduction of MIANs by metallic sodium and structurally characterize the reduced products. The comparison of the redox-sensitive bond parameters involving –N=C–C=O fragments in the crystals of sodium complexes toward corresponding neutral MIANs confirmed the reduced form of the ligand, namely C–C bond elongation as well as C–O and C–N bond shortening were observed for anion radicals compared to the neutral MIANs. The electrochemical reduction of MIANs at first-reduction-peak potential led to diamagnetic two-electron, one-proton addition products, which were structurally characterized. Electrochemically reduced products were tetrabutylammonium salts of protonated anionic MIAN. A similar but more essential trend of C–C bond elongation as well as C–O and C–N bond shortening in the initial –N=C–C=O fragment, compared to the neutral MIANs, was observed in the crystals. Thus, differences in the reduction products of chemical and electrochemical reactions were shown. The reduced products of both reactions were studied electrochemically. We found a more cathodic reduction peak on the CVs as well as a bathochromically shifted absorbance band in the UV/Vis spectra for **1**–**9**-NBu_4_ in comparison to corresponding values for sodium complexes **1**–**9**-Na. The results on two types of reduction products forming after chemical and electrochemical reductions could be helpful for the design of new catalytic systems or the prediction of the reactivity of MIAN complexes for different reactions.

## Data Availability

Data are contained within the article and the Appendix A.

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
