# Peer review of "Chemical and Electrochemical Reductions of Monoiminoacenaphthenes"

_ijms, 2023, doi:10.3390/ijms24108667_

Round 1

Reviewer 1 Report

In this work, Vera V. Khrizanforva and his/her colleagues provide a detailed analysis of the redox properties of MIANs and the one-electron reduction of MIANs with the aid of various electrochemical techniques. The authors computed the HOMO-LUMO gap, the data of the oxidation/reduction potentials and the electronic absorption spectra within DFT and TDDFT approaches. The systematic work points out the chemical and electrochemical reduction process of the two compounds shed light on the design of new catalytic systems and the prediction of the reactivity of MIAN complexes. I think the topic of this work is timely and the findings presented are very interesting. Overall the paper is well written, and I recommend the present publication of this work in IJMS.

Minor editing of English language required

Author Response

Dear Editor and Referees,

We thank you for your careful reading of the manuscript and fruitful revision. All corrections have been highlighted in yellow. English was polished during the text.

We hope that the revised manuscript would merit the highest standards of such a highly evaluated scientific forum as the International Journal of Molecular Science.

Yours sincerely,

Almaz Zagidullin,

Vera Khrizanforova

Reviewer 2 Report

The manuscript named "Differences between chemical and electrochemical reductions 3 of monoiminoacenaphthenes" reveals important insights into MIANs and their sodium complexes. I recommend the authors address the following comments.

1. The authors must do the UV-vis comparison of the sodium complex and MIANs.

2. HPLC analysis is required for the complexation

3. The title of the manuscript has not opted 

4. Calculate the number of electrons in electrochemical MIANs reductions and mention the method.

5. Derive the mechanism for electrochemical MIANs reductions.

Requires minor English corrections

Author Response

Dear Editor and Referees,

We thank you for your careful reading of the manuscript and fruitful revision. All corrections have been highlighted in yellow. English was polished during the text.

We hope that the revised manuscript would merit the highest standards of such a highly evaluated scientific forum as the International Journal of Molecular Science.

Yours sincerely,

Almaz Zagidullin,

Vera Khrizanforova

Responses to Referee 2:

The manuscript named "Differences between chemical and electrochemical reductions 3 of monoiminoacenaphthenes" reveals important insights into MIANs and their sodium complexes. I recommend the authors address the following comments.

Q1: The authors must do the UV-vis comparison of the sodium complex and MIANs.

R1: Thank you. We compared UV/vis spectra of the sodium complexes and neutral MIANs and added discussion in the text.

Q2: HPLC analysis is required for the complexation.

R2: Thank you. HPLC analysis is not suitable for MIAN complexes due to their air sensitivity.

Q3: The title of the manuscript has not opted.

R3: The title has been checked.

Q4: Calculate the number of electrons in electrochemical MIANs reductions and mention the method.

R4: The number of electrons required for electrochemical reductions of the MIANs at the first peak potential were calculated on the base of data-controlled potential electrolysis (CPE). The total charge passed during the CPE experiment (Q) is calculated by integrating the current and is related to the number of electrons transferred per molecule (n) and the number of moles of the reduced species initially present (N) through Faraday's law: Q = nFN. Following this, we determined the number of electrons corresponding to 2.

Q5: Derive the mechanism for electrochemical MIANs reductions.

R5: We added the proposed mechanism for the electrochemical reduction of MIANs in the text.

Reviewer 3 Report

The ms reports a study on the differences between chemical and electrochemical reductions of monoiminoacenaphthenes. Overall, the ms is well written with a good set of experimental data. Points to be addressed:

1) The abstract is very general. For example they stated that this and that were studied using this or that technique. I think the abstract should give an idea to the reader of the main findings;

2) Same as above for the conclusions. I would add more details in that;

3) it is not clear from the introduction whether this MIAN compounds are newly synthesized compounds or are they already been reported in literature? If so in what this contribution is new? This is not deduced in the introduction;

4) as in this study, electrochemistry is prevalent so I would add the experimental description of the electrochemical characterisation in the main text rather than in the ESI section;

Author Response

Dear Editor and Referees,

We thank you for your careful reading of the manuscript and fruitful revision. All corrections have been highlighted in yellow. English was polished during the text.

We hope that the revised manuscript would merit the highest standards of such a highly evaluated scientific forum as the International Journal of Molecular Science.

Yours sincerely,

Almaz Zagidullin,

Vera Khrizanforova

Responces to Referee 3:

The ms reports a study on the differences between chemical and electrochemical reductions of monoiminoacenaphthenes. Overall, the ms is well written with a good set of experimental data. Points to be addressed:

  • The abstract is very general. For example they stated that this and that were studied using this or that technique. I think the abstract should give an idea to the reader of the main findings;

R1: The abstract was rewritten and improved.

  • Same as above for the conclusions. I would add more details in that;

R2: Conclusions were also improved.

  • it is not clear from the introduction whether this MIAN compounds are newly synthesized compounds or are they already been reported in literature? If so in what this contribution is new? This is not deduced in the introduction;

R3: The neutral MIANs used in work are literature known. This sentence has been added in the text. However, there are no voltamperometric studies, as well as the chemical and electrochemical reductions, of MIANs in the literature. We first performed voltamperometric study of a series of substituted MIANs and evaluated HOMO/LUMO gaps by different ways. Firstly, chemical and electrochemical reduction of MIANs and following voltamperometric study of reduced forms was performed. The seven new reduced MIANs compounds were structurally characterized and discussed in the text. The found key bond distances in MIAN moiety make a valuable contribution to future research and could be helpful in the ligand and metal oxidation states in complexes

  • as in this study, electrochemistry is prevalent so I would add the experimental description of the electrochemical characterisation in the main text rather than in the ESI section;

R4: Thank you. The entire experimental part was placed in the ESI in order not to clutter up the main text of the article. Despite this, the electrochemical part is described in some detail and it will not be difficult to find the experiment features in the ESI part.

Round 2

Reviewer 2 Report

The authors have corrected required changes. The manuscript can be accepted.